# GenVidBench: A Challenging Benchmark for Detecting AI-Generated Video

## Abstract

The rapid advancement of video generation models has made it increasingly challenging to distinguish AI-generated videos from real ones. This issue underscores the urgent need for effective AI-generated video detectors to prevent the dissemination of false information through such videos. However, the development of high-performance generative video detectors is currently impeded by the lack of large-scale, high-quality datasets specifically designed for generative video detection. To this end, we introduce GenVidBench, a challenging AI-generated video detection dataset with several key advantages: 1) Cross Source and Cross Generator: The cross-generation source mitigates the interference of video content on the detection. The cross-generator ensures diversity in video attributes between the training and test sets, preventing them from being overly similar. 2) State-of-the-Art Video Generators: The dataset includes videos from 8 state-of-the-art AI video generators, ensuring that it covers the latest advancements in the field of video generation. 3) Rich Semantics: The videos in GenVidBench are analyzed from multiple dimensions and classified into various semantic categories based on their content. This classification ensures that the dataset is not only large but also diverse, aiding in the development of more generalized and effective detection models. We conduct a comprehensive evaluation of different advanced video generators and present a challenging setting. Additionally, we present rich experimental results including advanced video classification models as baselines. With the GenVidBench, researchers can efficiently develop and evaluate AI-generated video detection models.

## 1 Introduction

In recent years, video generation models like Sora have seen remarkable advancements (29; 5; 1), leading to a significant enhancement in the quality of AI-generated videos. The line between these realistic synthetic videos and real videos has become increasingly blurred, posing a variety of challenges, such as the spread of misinformation, damage to personal and corporate reputations, and an escalation in cybersecurity threats (3; 27). To address these risks, there is an urgent demand for the development of sophisticated AI-generated video detectors that can accurately identify and differentiate between real and fake videos. However, there is still a lack of challenging datasets to develop and evaluate AI-generated video detectors. To this end, we present, GenVidBench, a rich and challenging dataset for the development of AI-generated video detectors.

GenVidBench is an innovative dataset with cross-generation sources and cross-generators. It covers 8 types of videos generated by state-of-the-art video generators, such as Mora, MuseV, SVD, Pika, ensuring that the generated videos are of exceptional quality. GenVidBench contains two real video sources: Vript and HD-VG-130M. To make the dataset challenging, we construct 2 video pairs, each with the same text prompt/image. Specifically, we generate prompts and images based on the real video and use Image-to-Video (I2V) models and Text-to-Video (T2V) models to generate videos with the same source. Videos with the same generation source have more similar attributes, which makes it more difficult to distinguish real videos from fake videos. In GenVidBench, video pairs from the same source as the real video are used to form the test set, and video pairs from other sources are used to form the training set. Furthermore, the generators in the training set and the test set are different, so as to avoid the case in which videos generated by the same generator have the same attribute. Based on the above analysis, the task of cross-sources and cross-generators is extremely challenging.

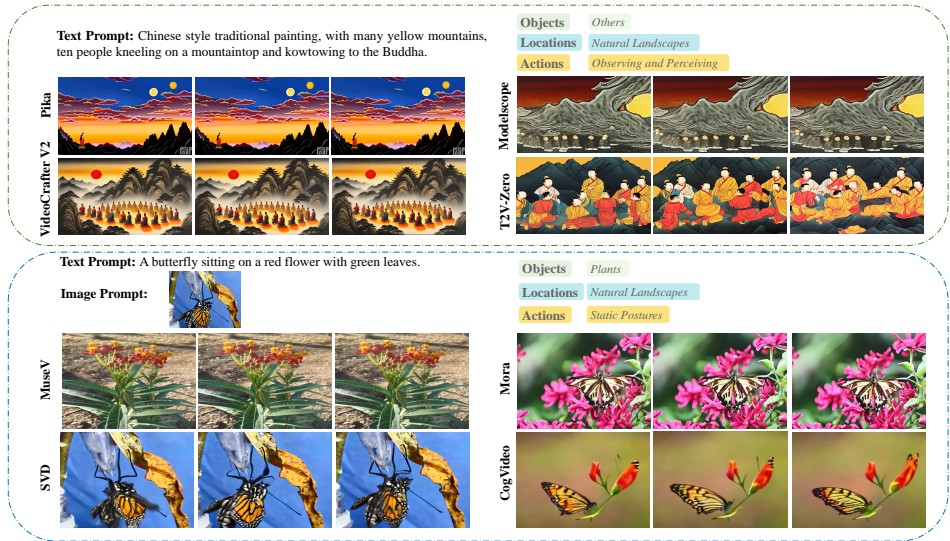

Figure 1: Introduction of the proposed GenVidBench dataset. The GenVidBench dataset not only contains the tags of real and fake videos, but also provides rich semantic content tags, such as object analogy, location, and action.

GenVidBench is the first 100k-level dataset that provides rich semantic content labels. Researchers can select corresponding types of videos as required to adapt to different application scenarios. We describe the content of the video in three dimensions: object category, action, and location. 1) Objects: the main character of the given video and decided the spatial major content. 2) Actions: a reflection of the temporal attributes. 3) Locations: an indicator of scenario complexity. As for Nature Landscapes, it may have a clean background, but for the Transportation scenario, the background may contain precision machines. Through analysis, we find that the object category affects the characteristics of the generated video. Therefore, a better training effect can be achieved by training a specific scenario by using a video of a corresponding category.

Comprehensive experimental results are presented to establish a solid foundation for researchers working on the development and assessment of AI-generated video detectors. A number of state-of-the-art video classifiers are evaluated on the GenVidBench dataset, including VideoSwin, UniformerV2, etc. The findings from our experiments underscore the considerable challenge inherent in cross-source and cross-generator tasks. Moreover, we conduct a detailed analysis of particularly challenging cases, employing semantic content labels to identify and filter the most challenging categories. The results from these difficult categories are also detailed, offering an expanded set of benchmarks to further assist researchers in the field.

## 2 RELATED WORK

### 2.1 AI-GENERATED CONTENT DETECTION DATASET

With the rapid developments of generative models, the requisite expertise and effort to generate content has been reduced. This fact has led to a growing focus on discriminating real items from AI-generated content and the construction of corresponding detection datasets. We categorize these detection datasets into 3 parts: AI-generated images, deepfake videos, AI-generated videos.

**AI-Generated Image Detection Dataset**. As a result of blossoming of diffusion models, AI-generated images also has become more realistic. GenImage (47) is a million-scale benchmark for detecting AI-generated images, which contains generated image pairs based on ImageNet (10) using various diffusion-based models and GAN-based models. Additionally, WildFake(16), ArtiFact (31), and DiffusionDB (40) also demonstrate the potential to provide a more comprehensive benchmark for fake image detection.

**Deepfake Video Detection Dataset**. There is a significant amount of research has focused on detecting deepfake videos (19; 44; 45; 9; 17) based on Deepfake datasets such as Deepfake Detection Challenge Dataset (DFDC) (11), Celeb-DF (24), FaceForensics++ (32), DeepFakeDetection (DFD) (14). These dataset are mainly use GANs, VAEs, or other swapping techniques to create fake videos.

**AI-Generated Video Detection Dataset**. Only a few research studies have focused on detecting purely generated videos on past years. Generated Video Dataset (GVD) (3) is constructed by 11,618 video samples yielded by 11 state-of-the-art generator models. GenVideo (7) is a large-scale AI-generated video detection dataset that collects videos from 10 different generated models for training, and videos from other 10 different generated models for testing. However, neither GVD nor GenVideo has the original prompt or images, video pair, semantic label, and cross source settings, which is shown in Table 1. As a result, their datasets have no way to avoid the problem of similar content in the training set and the test set, and there is no way to distinguish between different scenarios. Generated Video Forensics (GVF) (27) consists of videos pairs from 4 different text-to-video models using the same prompts extracted from real videos. As illustrated in Table 1, GVF contains prompts/images, video pairs, semantic labels. But it too small and only has 2.8k videos. The GVF does not have the cross source setting. The proposed GenVidBench is the first dataset with a scale of 100,000 containing semantic labels and prompts/images used to generate videos. Furthermore, GenVidBench performs cross-source setting of the training set and the test set, which is extremely challenging for fake video detection.

## 2.2 GENERATED VIDEO DETECTIONS

AI-generated videos have the potential to accelerate the dissemination of misinformation, prompting significant concern. In the past, much of the research has concentrated on detecting videos with synthetic faces (41). However, the content of the fake face video is single, which is greatly different from the real-world scenario. Due to the lack of large-scale high-quality AI-generated video datasets, there are few work on generate video detectors. Recent work on AI-generated video detection includes AIGDet (3), DeCoF (27). AIGDet (3) captures the forensic traces with a two-branch spatio-temporal convolutional neural network to improve detection accuracy. DeCoF (27) is based on the principle of video frame consistency to eliminates the impact of spatial artifacts. Beyond the specialized models dedicated to the detection of generated videos, several video classification models have demonstrated remarkable performance on this task, including VideoSwin (26) and UniFormer V2 (22). In this paper, we present the experimental results across various models on GenVidBench, which will provide a good research foundation for developers in related fields.

Table 1: An overview of fake video detection datasets. The proposed GenVidBench is the first dataset with a scale of 100,000 containing semantic labels and prompts/images used to generate videos. Furthermore, GenVidBench performs cross-source setting of the training set and the test set, which is extremely challenging for fake video detection.

| Dataset | Scale | Prompt/Image | Video Pairs | Semantic Label | Cross Source |
|---------|-------|--------------|-------------|----------------|--------------|
| GVD (3) | 11k | × | × | × | × |
| GVF (27) | 2.8k | √ | √ | √ | × |
| GenVideo (7) | 2271k | × | × | × | × |
| GenVidBench | 143k | √ | √ | √ | √ |

## 3 DATASET CONSTRUCTION

### 3.1 OVERVIEW OF GENVIDBENCH

The GenVidBench dataset primarily consists of real videos and fake videos shown in Table 2. The real videos are sourced from two existing datasets Vript (36) and HD-VG-130M (38), both of which include real videos and their corresponding descriptions. With an aim to balance content, we sample a total of 13800 videos from HD-VG-130M and 20131 videos from Vript. HD-VG-130M (38), established in May 2023, offers real videos at a resolution of 1280x720 and a frame rate of 30 FPS,

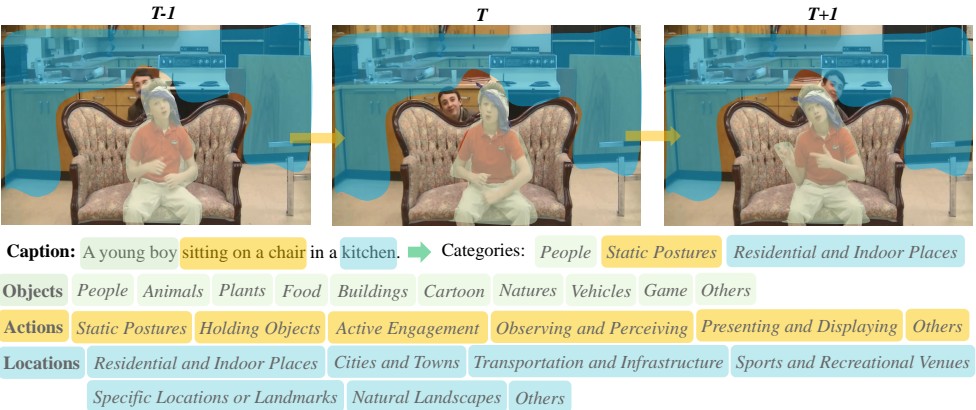

Figure 2: Different dimensions for content analysis. The final categories are shown in the bottom.

Table 2: Statistics of real and generated videos in the GenVidBench dataset. GenVidBench contains 8 subsets of fake videos generated by 8 state-of-the-art generators and 2 subsets of real videos. Two video pairs is divided based on the generation source such as the text prompt and the image.

| Video Source | Year | Pair | Type | Task | Resolution | FPS | Length | Numbers |
|---|---|---|---|---|---|---|---|---|
| Vript (36) | 2024.04 | - | Real | - | 1280x720 | 30 | 10 | 20131 |
| HD-VG-130M (38) | 2023.05 | Pair2 | Real | - | 1280x720 | 30 | ¿5 | 13800 |
| Pika (2) | 2022.05 | Pair1 | Fake | T2V | 1088x560 | 24 | 3 | 13500 |
| VideoCrafter2 (8) | 2024.01 | Pair1 | Fake | T2V | 512x320 | 10 | 1 | 13500 |
| Modelscope (37) | 2023.08 | Pair1 | Fake | T2V | 256x256 | 8 | 2 | 13500 |
| T2V-Zero (20) | 2023.03 | Pair1 | Fake | T2V | 512x512 | 4 | 2 | 13500 |
| MuseV (1) | 2024.03 | Pair2 | Fake | I2V | 1210x576 | 12 | 1 | 13800 |
| SVD (5) | 2023.11 | Pair2 | Fake | I2V | 1024x576 | 10 | 1 | 13800 |
| Mora (43) | 2024.03 | Pair2 | Fake | T2V | 1024x576 | 10 | 2 | 13800 |
| CogVideo (15) | 2022.05 | Pair2 | Fake | T2V | 480x480 | 4 | 3 | 13800 |
| Sum | - | - | - | - | - | - | - | 143131 |

with lengths ranging from 5 to 20 seconds. Vript (36), created in April 2024, also provides real videos at the same resolution and frame rate, with each video fixed at a length of 10 seconds.

The fake videos in the GenVidBench dataset are divided into two main categories, both generated from the same prompts or corresponding images. The first category includes videos from SVD (5) and MuseV (1), created by extracting frames from the HD-VG-130M dataset. Similarly, Mora (43) and CogVideo (15) videos are produced using prompts from the HD-VG-130M dataset. This process yields five pairs of videos with matching content, with each model contributing 13,800 videos to the dataset. The second category comprises videos from Pika, VideoCrafter2, Modelscope, and Text2Video-Zero. These videos are also generated from a uniform prompt and are sourced from the VidProM dataset (39), with each model providing 13,500 videos. By ensuring that the content of paired videos is identical, GenVidBench prevent AI video detectors from distinguishing between real and fake videos based on content alone, thereby increasing the dataset's challenge.

Here are the specifications for the fake videos generated by each model: Pika produces fake videos at a resolution of 1088×560 and a frame rate of 24 FPS. VideoCrafter2 generates fake videos at a resolution of 512×320 and a frame rate of 10 FPS. Modelscope creates fake videos at a resolution of 256×256 and a frame rate of 8 FPS. Text2Video-Zero generates fake videos with a resolution of 512×512 and a frame rate of 4 FPS. Musev generates fake videos with a resolution of 1210×576 and a frame rate of 12 FPS. SVD creates fake videos at a resolution of 1024×576 and a frame rate of 10 FPS. Mora produces fake videos with a resolution of 1024×576 and a frame rate of 10 FPS. CogVideo generates videos with a resolution of 480×480 and a frame rate of 4 FPS. Most videos are 1-2 seconds long, except Pika, which has 3 seconds.

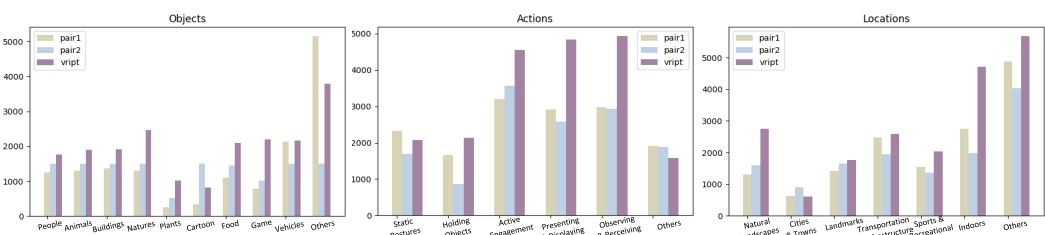

Figure 3: Data distribution over categories at different levels.

## 3.2 CONTENT ANALYSIS

To ensure the realism of the fake videos, we select the most state-of-the-art generators. MuseV (1), Mora (43), and VideoCrafter2 (8) are the newest generators, introduced in 2024. SVD (5) and Modelscope (37), proposed in November 2023 and August 2023, respectively, are also considered at the forefront of high-performance methods. Additionally, Pika (2), CogVideo (15), and Text2Video-Zero (20) are recognized for their excellent performance in video generation. To facilitate the development of more generalized and effective detection models, we categorized the available captions across multiple dimensions and extracted a balanced set of categories based on a primary dimension. As depicted in Fig. 2, we initially divided captions into three key levels: objects, actions, and locations, as these three elements can form a complete sentence or story with rich semantics.

1) **Objects**. The main subject of the given video, determining the primary spatial content. 2) **Actions**. Reflect the temporal attributes of the video. For instance, categorizing the action as Static Postures implies that the main subject of the video will not exhibit rapid movement. If the action is categorized as Presenting and Displaying, the main subject may engage in walking around objects or performing other bodily movements. 3) **Locations**. This indicates the complexity of the scenario. For example, a Nature Landscape may have a clean background, whereas a Transportation scenario might include intricate machinery in the background.

For each level, taking the object level as an example, we sampled 10% of the entire caption set and utilized Large Language Models (LLMs) to extract the subjects. We then aggregated these subjects into more abstract categories. The final categories were constrained to no more than ten options to construct a classification tree. Based on the classification tree, we can effortlessly classify captions and obtain the definitive class-labels for each caption.

We selected objects as our primary dimension for selecting and generating video pairs, ensuring that our benchmark is balanced and semantically rich in the spatial dimension. Fig. 3 illustrates the distribution of various caption sets across multiple dimensions. Fig. 4 presents the distribution of the entire benchmark. These figures demonstrate that our benchmark encompasses a vast array of data, covering extensive category ranges in both spatial and temporal aspects.

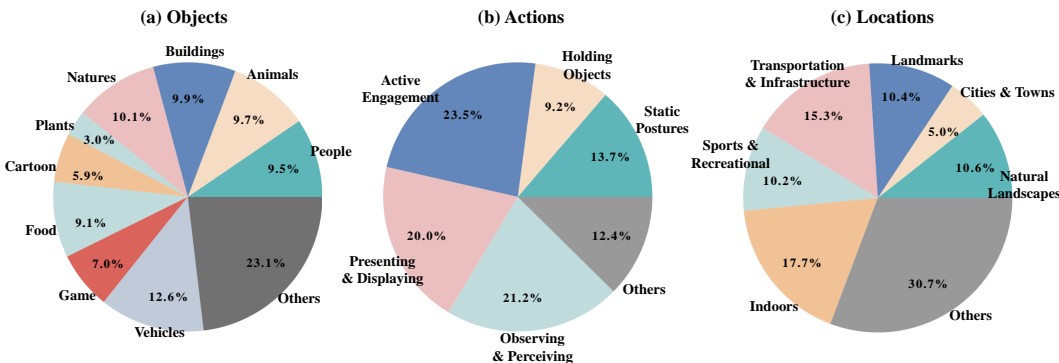

Figure 4: Data distribution of our benchmark.

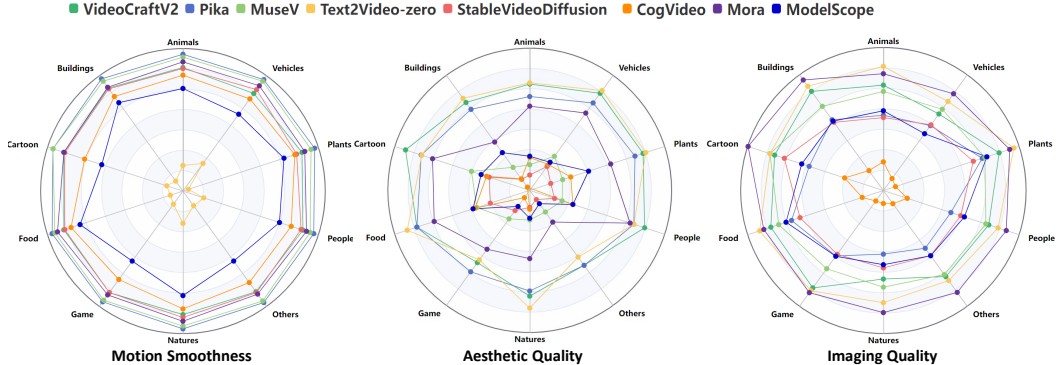

Figure 5: The evaluation results of generated videos across objects content level.

## 3.3 EVALUATION OF VARIOUS VIDEO GENERATORS

We selected eight representative open-source generators to build the benchmark, including T2V and I2V models with various generation paradigms. Fig. 5 shows the performance of videos generated by the eight models on different attributes based on VBench (18), where a plot closer to the outermost circle indicates higher quality. Motion smoothness serves as a temporal quality metric, while aesthetic and imaging qualities are utilized for frame-wise quality assessments. Aesthetic quality captures the visual appeal, including layout, color richness and harmony, photo-realism, naturalness, and artistic attributes of the video frames. Imaging quality pertains to the level of distortion in the video frames, such as blurriness.

Obviously, each generator excels and falls short in various aspects. No single generator is significantly superior across all categories on the three dimension. Text2Video-zero exhibits poor motion smoothness but delivers one of the most impressive impacts on frame-wise quality. Conversely, Pika, which leads in temporal quality, does not stand out in terms of aesthetic and imaging qualities. These distinct strengths and weaknesses ensure that our benchmark can effectively evaluate the performance of detectors when confronted with a variety of unseen video generators.

Beyond the generators themselves, Fig. 5 reveals interesting insights. In terms of imaging quality, achieving high standards is particularly challenging for Natural scenes and Vehicles across all generators. However, in the temporal dimension, they generally demonstrate superior performance. In contrast, Cartoons show lower motion smoothness but excel in frame-wise quality. This suggests that different semantic contents can significantly influence the attributes of the generated videos. When generation sources share similar content, the resulting video attributes tend to align more closely, complicating the differentiation between such videos. Besides, these observations also indicates optimization directions for generators customized to specific content types.

Table 3: Results of cross-validation on different training and testing subsets using Swin-tiny.

| Training Set | Test Set | | | | | | | | Mean |
|---|---|---|---|---|---|---|---|---|---|
| | Pika | VC2 | MS | T2VZ | MuseV | SVD | Cogvideo | Mora | |
| Pika | 99.76 | 95.14 | 54.91 | 55.17 | 65.63 | 54.66 | 69.86 | 53.63 | 64.68 |
| VC2 | 66.75 | 99.9 | 84.37 | 73.03 | 66.09 | 54.86 | 59.36 | 69.66 | 68.60 |
| MS | 50.6 | 51.11 | 99.89 | 52.68 | 59.68 | 54.37 | 41.03 | 49.11 | 51.26 |
| T2VZ | 50.56 | 51.98 | 55.46 | 99.69 | 60.28 | 55.06 | 43.43 | 50.66 | 52.51 |
| MuseV | 70.59 | 66.34 | 54.06 | 55.8 | 97.57 | 62.55 | 73.54 | 53.14 | 62.39 |
| SVD | 50.6 | 61.63 | 62.37 | 65.99 | 94.27 | 99.3 | 80.67 | 81.58 | 72.83 |
| Cogvideo | 51.07 | 60.84 | 60.36 | 68.07 | 60.64 | 57.61 | 99.83 | 52.55 | 58.51 |
| Mora | 51.84 | 76.14 | 62.12 | 62.48 | 60.96 | 86.17 | 49.84 | 99.69 | 64.40 |

### 3.4 CROSS-SOURCE AND CROSS-GENERATOR TASK

We first evaluate the performance of the detector when trained and tested on videos generated by the same generator. Our evaluation utilizes the widely recognized VideoSwin-tiny model (26). The proposed GenVidBench dataset can be divided into eight distinct subsets, each corresponding to a specific generator. Within each subset, we further divide the data into training and testing sets. As shown in Table 3, training and testing within each subset consistently yield accuracy exceeding 97.6%. Impressively, the VideoCrafter2 and Modelscope subsets achieve an exceptional accuracy of 99.9%. Drawing from this observation, detecting a video synthesized by a specific generator is relatively straightforward.

However, this approach is inherently generator-dependent and may fail to produce satisfactory results when dealing with unknown generators. It can be observed that a substantial performance degradation when training and testing are conducted using different generators. For instance, the accuracy of VideoSwin-tiny drops to 54.66% when it is trained on Pika and tested on SVD. In practical scenarios, the identity of the generator is often undisclosed. In this study, our objective is to Effectively evaluate the generalization performance of detectors, specifically its ability to discern between real and fake videos without reliance on the specific generator's identity. To achieve this, the cross-generator video classification task is introduced, aimed at test the detector's robustness and adaptability across a variety of video sources, thereby enhancing its applicability in diverse and unpredictable real-world scenarios.

Table 4: Comparison of the results when the training set and the test set are the same generation source and different generation source based on VideoSwin-tiny. The results show that testing on different generation sources is more challenging.

| Pair1 | Same Source | Different Source | Pair2 | Same Source | Different Source |
|---|---|---|---|---|---|
| Pika | **68.41** | 60.95 | Musev | **63.08** | 61.70 |
| VC2 | **74.72** | 62.49 | SVD | **85.51** | 60.15 |
| MS | **51.46** | 51.05 | Cogvideo | 56.93 | **60.09** |
| T2V-Z | **52.67** | 52.36 | Mora | **65.66** | 63.15 |
| Mean | **61.81** | 56.71 | Mean | **67.79** | 61.27 |

Furthermore, we analyze the impact of the generation source, such as text prompts and images, on the classification of true and false videos. The four models Pika (2) , VideoCrafter2 (8), Modelscope (37), and Text2Video-Zero (20) use a same text prompt to generate a video, and these videos are the first video pair. The MuseV (1) and SVD (5) models are based on the same image for video generation, while CogVideo and Mora rely on the same text prompt, forming our second set of video pairs. As shown in Table 4, training with videos from the same video source tends to yield superior test results. For example, in the first video pair, the Pika achieves an accuracy of 68.41% when test on the same source video, but only an accuracy of 60.95% on the video of other sources. In the second video pair, SVD achieves an accuracy of 85.51% when training with videos from the same source, yet this drops to 60.15% when tested on videos originating from different sources. Moreover, the average accuracy of testing with the same source videos exceeds that of testing with non-same source videos across both pairs. These findings underscore the the dependency of detectors on the uniformity of the generation source. To improve the robustness and generalization of the detector, we propose a challenging dataset and design a cross-source true-false video classification task. Videos from Pair1 and Vript are used as the training set, and videos from Pair2 and HD-VG-130M are used as the test set. The videos in the test set all have the same generation source, which makes it more difficult to distinguish between real and fake videos.

### 3.5 SCENARIO-SPECIFIC TASK

As mentioned in the section 3.2 and 3.3, we conduct a comprehensive content analysis of all captions and videos to obtain corresponding category labels and quality metrics for temporal and spatial domain generation. This process enables researchers to extract datasets customized to their specific research interests. For instance, they can extract People scenes to evaluate the logical of human body generation, extract Presenting scenes to study motion deformation, and extract scenes with low

temporal continuity to study the deformation of each frame. Furthermore, based on these extracted dataset we can evaluate the impact of these attributes on detector, such as the relationship of motion deformation and detection accuracy.

# 4 EXPERIMENTAL ANALYSIS

## 4.1 IMPLEMENTATION DETAILS

We sampled 8 frames from each video as input, and resized each image to $224 \times 224$. Batch size is 8. The default learning rate of each method in mmaction2 is used. Data augmentation methods such as random flipping and cropping are also used. The other training settings are the default configuration of MMAction2.

## 4.2 RESULT ON CROSS-SOURCE AND CROSS-GENERATOR TASK

Table 5: Results of various state-of-the-art methods trained on cross-source and cross-generator task. The superior performance metric does not exceed 79.90%, indicating that this task presents considerable challenges.

| Method | Type | MuseV | SVD | CogVideo | Mora | HD-VG | Top1 |
|---|---|---|---|---|---|---|---|
| SlowFast (13) | CNN | 12.25 | 12.68 | 38.34 | 45.93 | 93.63 | 41.66 |
| F3Net (30) | CNN | 37.43 | 37.27 | 36.46 | 39.59 | 52.76 | 42.52 |
| I3D (6) | CNN | 8.15 | 8.29 | 60.11 | 59.24 | 93.99 | 49.23 |
| CFV2 (28) | CNN | 86.26 | 86.53 | 10.10 | 16.90 | 88.40 | 60.53 |
| TPN (42) | CNN | 37.86 | 8.79 | 68.25 | 90.04 | 97.34 | 61.52 |
| TIN (33) | CNN | 33.78 | 21.47 | 81.59 | 79.44 | 97.88 | 63.97 |
| TRN (46) | CNN | 38.92 | 26.64 | 91.34 | 93.98 | 93.97 | 71.26 |
| TSM (25) | CNN | 70.37 | 54.70 | 78.46 | 70.37 | 96.76 | 76.40 |
| X3D (12) | CNN | **92.39** | 37.27 | 65.72 | 49.60 | 97.51 | 77.09 |
| UniFormer V2 (22) | Transformer | 20.05 | 14.81 | 45.21 | **99.21** | 96.89 | 57.55 |
| TimeSformer (4) | Transformer | 73.14 | 20.17 | 74.80 | 39.40 | 92.32 | 64.28 |
| VideoSwin (26) | Transformer | 62.29 | 8.01 | **91.82** | 45.83 | **99.29** | 67.27 |
| MViT V2 (23) | Transformer | 76.34 | **98.29** | 47.50 | 96.62 | 97.58 | **79.90** |

To furnish researchers with robust benchmarks for both evaluation and advancement, we have conducted a comprehensive performance assessment of several state-of-the-art video classification models on the GenVidBench dataset. These models include SlowFast (13), F3Net (30), I3D (6), CFV2 (28), TPN (42), TIN (33), TRN (46), TSM (25), X3D (12), UniFormer V2 (22), TimeS-former (4), VideoSwin (26) and MViT V2 (23). These methods implemented based on MMAction2 and are trained by using the default configuration provided by MMAction2.

As shown in Table 5, MViT V2 achieves the best accuracy, with a Top1 of 79.90%. The performance difference between the CNN model and the Transformer model is not large in the AI-generated video detection task. X3D also achieves 77.09% accuracy as a CNN model. Additionally, TRN and TSM have also demonstrated commendable classification outcomes, with Top1 accuracies of 71.26% and 76.40%, respectively. Conversely, I3D lags behind with a Top1 accuracy of only 49.23%. SlowFast achieves the poorest performance, attaining a mere 39.69% in Top1 accuracy. The above experimental results show that the cross-source and cross-generator task is indeed challenging, indicating that there remains substantial room for improvement in the field of generative video detection.

Besides, we can find that real videos are more easily distinguished from fake videos in AI-generated video detection task, with most classification accuracy greater than 88.40%. Fake videos generated by SVD are the hardest to classify correctly, which proves that its generation performance is the best. Except for MVit V2 and CapsuleForensicsV2, the classification accuracy of other models is less than 54.70%. Fake videos generated by CogVideo are the easiest to classify, due to their poor temporal continuity.

### 4.3 COMPARISON WITH ADVANCED DATASETS

We have contemplated the possibility of conducting a comparative categorical analysis on existing datasets, but due to the lack of publicly available datasets in some papers, we are unable to conduct a more in-depth analysis of the data. The Deepfake benchmarks only contain face videos, which lack diversity. The AI-Generated Video Detection Datasets, including the Generated Video Dataset (GVD) and Generated Video Forensics (GVF), have not released their datasets, thus we only use original papers for analysis. Notably, the original papers of GVD does not include a categorical analysis of the content. GVF's categorization of the spatial domain into people, animals, plants, food, vehicles, buildings, artifacts, scenery, and illustrations is not only less extensive than our own but also lacks semantic clarity. Furthermore, GVF's overall dataset volume is significantly smaller than ours.

To compare the challenges of different datasets, we have conducted a comparative analysis of the performance achieved by the SlowFast, I3D, and F3Net models across a spectrum of datasets, as detailed in Table 6. We follow the results presents in previous work (27; 7) and add the new results trained on our benchmark.

It can be found that SlowFast, I3D, and F3Net perform much better on other datasets than on our GenVidBench. SlowFast is tested on multiple datasets and its performance on GVF dataset is low at 60.95%. However, its performance on GenVidBench is only 41.66%. I3D achieves an accuracy of 61.88% on the GVF dataset, but only gets 49.23% on our proposed GenVidBench. F3Net achieves an accuracy of 51.83% on the GenVideo dataset, but only attains 42.52% on our proposed GenVidBench. This shows that our dataset is more challenging and can provide more room for improvement of the detector.

Table 6: Performance comparison of detectors on different datasets.

| Dataset | SlowFast | I3D | F3Net |
|---|---|---|---|
| DeepFakes (14) | 97.53 | - | - |
| Face2Face (35) | 94.93 | - | - |
| FaceSwap (21) | 95.01 | - | - |
| NeuralTextures (34) | 82.55 | - | - |
| GVF (27) | 60.95 | 61.88 | - |
| GenVideo (7) | - | - | 51.83 |
| GenVidBench(Ours) | **41.66** | **49.23** | **42.52** |

### 4.4 HARD CASE ANALYSIS

To further explore the detection results, we present a hard case analysis based on the experimental results of VideoSwin-tiny. We define the generated videos for which the detector predicts a very low probability ($< 3\%$) as our hard cases. The result is shown in Table. 7.

Table 7: The proportion of hard cases on different categories in different generators. The Plants class is the most likely to be misclassified, while the Cartoons class is the easiest to distinguish.

| Method | People | Animals | Buildings | Natures | Plants | Cartoon | Food | Game | Vehicles | Mean |
|---|---|---|---|---|---|---|---|---|---|---|
| SVD | 0.757 | 0.728 | 0.777 | 0.736 | 0.739 | 0.537 | 0.769 | 0.732 | **0.795** | 0.739 |
| MuseV | 0.156 | 0.152 | 0.16 | 0.164 | **0.22** | 0.098 | 0.117 | 0.189 | 0.129 | 0.16 |
| Mora | **0.226** | 0.153 | 0.196 | 0.21 | 0.205 | 0.157 | 0.171 | 0.156 | 0.215 | 0.198 |
| CogVideo | 0.014 | 0.008 | 0.01 | 0.023 | 0.012 | 0.004 | 0.01 | **0.036** | 0.013 | 0.017 |
| Mean | 0.307 | 0.277 | 0.306 | 0.303 | **0.318** | 0.209 | 0.286 | 0.295 | 0.308 | 0.297 |

Although the detector exhibits remarkable and consistent performance in all test videos, it shows significant differences in performance when facing cross-generator scenarios. For example, from the Table. 7, we can see that SVD presents more challenging scenarios for the detector, while CogVideo has almost no server ambiguous scenarios.

Moreover, as shown in Fig.6, for different spatial major contents, the proportion of hard cases also varies across different generators. For instance, vehicles are the most difficult to determine the

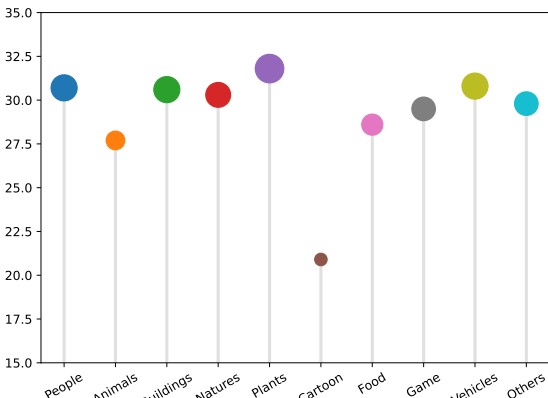

Figure 6: The hard-case occupation percentage of different categories.

authenticity for SVD, but for Musev, scenes involving vehicles are relatively easier to distinguish. This may reflect the expertise of each generator in generating subjects and also indicate the future optimization directions for the generators. Among all categories, the plants category emerges as the most difficult scenario, prompting us to focus on plant videos as our challenging task.

### 4.5 RESULT ON SCENARIO-SPECIFIC TASK

Based on the analysis of hard examples, we select the Plants class and give the experimental results of various models on this class. As illustrated in Table 8, TimeSformer achieves the best accuracy, with a Top1 of 75.09%. Additionally, TPN and UniFormerV2 also demonstrate commendable classification Results, with Top1 accuracies of 64.24% and 64.76%, respectively. Conversely, SlowFast lags behind with a Top1 accuracy of only 55.30%. VideoSwin achieves the poorest performance, attaining a mere 52.86% in Top1 accuracy. Experimental results also show that SVD has the lowest classification accuracy, which proves that its generation performance is the best. The above experimental results show that the classification performance of the model in a single scene is different from that in all scenes, so it is necessary to generate video detection for different scenes. The rich semantic tags provided by GenVidBench can help analyze each scenario and provide more development ideas for developers.

Table 8: Results of various state-of-the-art methods trained on plants class.

| Method | MuseV | SVD | CogVideo | Mora | HD-VG-130M | Mean Top1 |
|---|---|---|---|---|---|---|
| I3D (6) | 39.18 | 23.27 | 91.98 | 78.38 | 78.42 | 62.15 |
| SlowFast (13) | 81.63 | 29.80 | 75.31 | 19.31 | 73.03 | 55.30 |
| TPN (42) | 43.67 | 20.00 | 85.80 | 86.87 | 94.61 | 64.24 |
| TimeSformer (4) | 77.96 | 29.80 | 96.30 | 93.44 | 87.14 | 75.09 |
| VideoSwin (26) | 57.96 | 7.35 | 92.59 | 47.88 | 98.76 | 52.86 |
| UniFormerV2 (22) | 13.88 | 7.76 | 41.98 | 95.75 | 97.93 | 64.76 |

## 5 CONCLUSION

In this paper, we introduce GenVidBench, an innovative dataset designed for the detection of AI-generated videos, characterized by its cross-source and cross-generator settings. GenVidBench is the first 100,000-level dataset that provides rich semantic content labels. Comprehensive experimental results are presented to establish a solid foundation for researchers working on the development and evaluation of AI-generated video detectors. Moreover, we conduct a detailed analysis of particularly challenging cases, employing semantic content labels to identify and filter the most challenging categories. The results from these difficult categories are also detailed, offering an expanded set of benchmarks to further assist researchers in various scenarios.

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
