# OpenReview forum: "GenVidBench: A Challenging Benchmark for Detecting AI-Generated Video"
_ICLR.cc/2025/Conference — ICLR 2025 Conference Withdrawn Submission_

### Official Review · Reviewer_gGGM · 2024-10-29

**Soundness:** 2
**Presentation:** 2
**Contribution:** 2
**Rating:** 3
**Confidence:** 4

**Summary:**

This paper introduces GenVidBench, a challenging benchmark dataset for detecting AI-generated videos. GenVidBench addresses the lack of large-scale, high-quality datasets specifically designed for generative video detection by including videos from eight state-of-the-art AI video generators, covering the latest advancements in video generation. The dataset features cross-source and cross-generator attributes to enhance diversity and prevent overfitting between training and test sets. It also offers rich semantic classifications, aiding in the development of more generalized and effective detection models. The authors conduct comprehensive evaluations using advanced video classification models as baselines, providing extensive experimental results to facilitate the development and evaluation of AI-generated video detection models.

**Strengths:**

The paper provides a rich analysis, thoroughly detailing the construction of the dataset and offering extensive specifics. It includes numerous details about the dataset, enhancing its transparency and usefulness for future research.

**Weaknesses:**

- The dataset does not provide an anonymous link for complete inspection; only a few examples are given, making it difficult to assess its completeness and quality.

- The dataset's size and its contribution to generative video detection models are not significantly superior to existing datasets, casting doubt on its overall contribution.

- While the dataset claims to have unique settings in terms of original prompts/images, video pairs, semantic labels, and cross-source attributes, with corresponding analyses, similar configurations exist in other datasets. Some of these contributions are minor (e,g,, publishing the prompts). The paper's separate analysis of these aspects does not yield particularly novel conclusions.

**Questions:**

- Section 3.2 mentions that the dataset considers factors such as object, action, and location. Do these factors also affect detection accuracy?

- In Table 3, the detection rates for the same model are high (close to 100%). Why is the accuracy for the same source in Table 4 not as high?

- In Section 4.3, it is mentioned that the dataset is more challenging and therefore more important. However, could this be due to lower data quality? Has any quality assessment been conducted?

---

### Official Review · Reviewer_RZp6 · 2024-11-02

**Soundness:** 3
**Presentation:** 3
**Contribution:** 2
**Rating:** 5
**Confidence:** 5

**Summary:**

This paper introduces GenVidBench, a challenging benchmark dataset for the detection of AI-generated videos. With the rapid advancement of video generation models, distinguishing between AI-generated and real videos has become increasingly difficult, highlighting the urgent need for effective AI-generated video detectors to prevent the spread of misinformation through such videos.

**Strengths:**

A challenging benchmark: 1) Cross Source and Cross Generator: Mitigating interference from video content and ensuring diversity in video attributes between training and test sets; 2) State-of-the-Art Video Generators: Including videos from 8 cutting-edge AI video generators, reflecting the latest advancements in the field; 3) Rich Semantics: Videos are analyzed and classified into various semantic categories based on content, ensuring the dataset is not only large but also diverse, aiding in the development of more generalized and effective detection models.

**Weaknesses:**

W1: The authors did not attempt to propose a baseline method based on experimental findings but merely utilized existing methods for experimentation.

W2: The paper mentions that the Plants category is the most likely to be misclassified, suggesting a potential bias towards certain semantic categories. This could mean that the dataset is not equally challenging across all categories, which might affect the robustness of detection models when applied to a diverse range of content.

W3: Most videos in GenVidBench are relatively short (1-2 seconds) and have varying resolutions. This could limit the dataset's ability to train detectors that are effective on longer videos or those with different resolutions, which are common in real-world applications.

**Questions:**

Q1: It appears that the model trained with SVD has the best generalization ability. Why does the generalization ability of high-quality data (pika, Cogvideo) turn out to be worse instead?

---

### Official Review · Reviewer_aCF4 · 2024-11-03

**Soundness:** 2
**Presentation:** 3
**Contribution:** 1
**Rating:** 3
**Confidence:** 4

**Summary:**

This paper introduces a novel dataset designed to facilitate the detection of AI-generated videos. The dataset is organized into cross-source and cross-generator settings, containing over 100,000 samples along with semantic content labels. The authors conducted extensive experiments across multiple dimensions, evaluating various video generation algorithms and detectors for AI-generated content. The paper also provides a detailed analysis of challenging cases and presents quantitative results.

**Strengths:**

1. Detecting AI-generated videos is an important research direction. The dataset and quantitative experiments presented in this paper contribute valuable resources and insights to the field.
2. The experimental design is thorough, and the paper is well-organized, making it clear and accessible.

**Weaknesses:**

1. Video generation is a rapidly advancing field, but as shown in Table 2, the methods evaluated in this paper are relatively outdated—some are at least six months old, with others dating back two years. Notably, more recent models, such as Sora, Gen-3, KlingAI, and HailuoAI, are absent from the evaluation, which weakens the relevance and robustness of the conclusions.

2. Although the primary aim of this work is to establish benchmarks for AI-generated video detection, there appears to be some overlap with AI video generation evaluation, as seen in Figure 5. This mixed focus detracts from the paper’s clarity. Additionally, recent studies on AI video generation evaluation are not cited, which limits the context for interpreting the results.

3. The categorization and distribution of data within the dataset are not entirely convincing. For example, in Figure 4, “cartoon” is categorized under "Objects" alongside categories like People and Animals. However, "cartoon" is more accurately a style, as there can be cartoon versions of people, animals, etc. Moreover, Figure 4 shows only the distribution within the proposed dataset without demonstrating whether this distribution accurately reflects real-world distributions relevant to the task.

4. Several important factors are overlooked in the analysis. First, real and generated videos often differ significantly in length; as the paper notes, generated videos are mostly 1–2 seconds long, whereas real videos typically have much longer durations. This discrepancy in length distribution could impact the study's conclusions. Second, the length of the input prompts is crucial to video generation quality, yet this factor is not examined in the paper.

5. The paper claims the benchmark is challenging, particularly noting that videos generated by SVD are the hardest to classify accurately (line 428). However, SVD is outdated and falls well behind state-of-the-art video generation methods, which undermines the strength of this claim.

**Questions:**

n/a

---

### Official Review · Reviewer_TJdC · 2024-11-04

**Soundness:** 2
**Presentation:** 2
**Contribution:** 2
**Rating:** 5
**Confidence:** 4

**Summary:**

This paper introduces GenVidBench, a comprehensive dataset designed to advance AI-generated video detection. GenVidBench includes videos from eight state-of-the-art generators, spans diverse semantic categories, and incorporates cross-source and cross-generator features to improve detection robustness. By providing challenging benchmarks and baseline results, GenVidBench aims to accelerate research in distinguishing AI-generated content from real video, addressing a critical gap in generative video detection.

**Strengths:**

--> The main contribution of this work is the proposed GenVidBench dataset which contains videos from 8 state-of-the-art video generators and 2 real video sources. Incorporating videos from multiple generators and adding semantic attributes to the videos to ensure diversity helps with the quality of the dataset.

--> The paper evaluates several state-of-the-art classifiers on GenVidBench and provides detailed analyses of challenging cases, establishing robust benchmarks and offering insights that can guide future research in developing more accurate detection models.

**Weaknesses:**

--> The cross-source aspect of the dataset is not very well described. It is unclear what the difference between the cross-source and cross-generator is.

--> Collection of the dataset and running baseline models on the collected dataset is an insufficient contribution for the work to be accepted in the main track of the conference. Proposing an approach to address fake video detection would have to be the novelty of the paper.

Minor
--> Formatting issue in Table 2, inverted ? in row 2.
--> The citations in the paper do not follow the ICLR template.

**Questions:**

why consider only Vript and HD-VG-130M datasets to sample real videos for your dataset? Any video dataset with captions could have been used.

---

### Note · Authors · 2024-11-14

I have read and agree with the venue's withdrawal policy on behalf of myself and my co-authors.